# Occupational Radiation Exposure and Validity of National Dosimetry Registry among Korean Interventional Radiologists

**DOI:** 10.3390/ijerph18084195

**Published:** 2021-04-15

**Authors:** Seulki Ko, Kwang Pyo Kim, Sung Bum Cho, Ye Jin Bang, Yae Won Ha, Won Jin Lee

**Affiliations:** 1Department of Preventive Medicine, Korea University College of Medicine, Seoul 02841, Korea; kowis@korea.ac.kr (S.K.); byj310@korea.ac.kr (Y.J.B.); lucyha92@korea.ac.kr (Y.W.H.); 2Graduate School of Public Health, Korea University, Seoul 02841, Korea; 3Department of Nuclear Engineering, Kyung Hee University, Gyeonggi-do 02447, Korea; kpkim@khu.ac.kr; 4Department of Radiology, Korea University Anam Hospital, Korea University College of Medicine, Seoul 02841, Korea; angler@korea.ac.kr

**Keywords:** interventional radiology, ionizing radiation, thermoluminescent dosimetry, national dose registry, epidemiologic methods

## Abstract

The national dose registry (NDR) contains essential information to help protect radiation workers from radiation-related health risks and to facilitate epidemiological studies. However, direct validation of the reported doses has not been considered. We investigated the validity of the NDR with a personal dosimeter monitoring conducted among Korean interventional radiologists. Among the 56 interventional radiologists, NDR quarterly doses were compared with actively monitored personal thermoluminescent dosimeter (TLD) doses as standard measures of validation. We conducted analyses with participants categorized according to compliance with TLD badge-wearing policies. A correlation between actively monitored doses and NDR doses was low (Spearman ρ = 0.06), and the mean actively monitored dose was significantly higher than the mean NDR dose (mean difference 0.98 mSv) in all participants. However, interventional radiologists who wore badges irregularly showed a large difference between actively monitored doses and NDR doses (mean difference 2.39 mSv), and participants who wore badges regularly showed no apparent difference between actively monitored doses and NDR doses (mean difference 0.26 mSv). This study indicated that NDR data underestimate the actual occupational radiation exposure, and the validity of these data varies according to compliance with badge-wearing policies. Considerable attention is required to interpret and utilize NDR data based on radiation workers’ compliance with badge-wearing policies.

## 1. Introduction

The dose registry system was initiated with the principal purpose of implementing regulatory controls and information collection for statistical analysis and reports [1]. Many countries maintain a national dose registry (NDR) and it is essential to protect radiation workers from radiation-related health risks. To monitor the exposure level of individual radiation workers, dose data obtained from individual dosimeters are collected periodically. NDR data are also important for epidemiological studies estimating the effects of low dose and low dose rate ionizing radiation [2]. These data have been used to evaluate or improve the effectiveness of safety programs [3,4].

The accuracy of reported dose data should be the essential premise of usage dose registries; epidemiological studies have emphasized uncertainties in radiation doses in particular [5,6]. Various sources and types of errors on dosimetry have been investigated, and previous studies were focused on the exposure assessment at a group level, relevant laboratory technology, and responsibility of dosimetry [6,7,8]. However, evaluation of the validity of dose data is limited and previous studies have indirectly investigated this aspect based on biodosimetry [9,10]. To our knowledge, the direct validation of reported doses, i.e., of the badge doses themselves, has not been considered for radiation workers. Information on validity of NDR data and factors influencing validity could contribute to reduce uncertainty of the reported dose data.

Medical workers who perform fluoroscopically guided interventional procedures are exposed to substantial doses of radiation compared to other medical radiation workers working with conventional radiography [11]. The higher occupational radiation exposure is related to the introduction of more complex interventional procedures [12], as well as the close proximity to the radiation sources such as X-ray tubes and patients [13]. Moreover, according to the increase of these procedures, concerns are mounting about health risks from radiation exposure to interventional medical workers [14]. Several studies have reported risks of cancer, cataract, and other health effects among interventional medical workers [15].

To investigate the occupational exposure status and the validity of NDR data among interventional medical workers, therefore, we conducted personal monitoring and compared the actively monitored doses and the Korea Centers for Disease Control and Prevention (KCDC) NDR-reported doses.

## 2. Materials and Methods

### 2.1. Study Population

We conducted a field survey of the interventional medical workers study in South Korea in 2017; the study protocol has been described previously in detail [16]. The study included a detailed questionnaire and personal dosimeter-monitoring of 56 interventional radiologists. The participants were invited through nationwide branches of the Korean Society of Interventional Radiology (KSIR), and 21.5% (*n* = 56) of the active KSIR members voluntarily participated with informed written consent, from 40 hospitals countrywide. A total of 56 participants completed the personal dosimeter monitoring program, and all were included in the statistical analyses (Figure 1). The Institutional Review Board of Korea University approved the protocol of this study (KU-IRB-17-36-A-2).

### 2.2. Actively Monitored Personal Thermoluminescent Dosimeter Doses

The occupational radiation exposure of interventional radiologists was measured with close monitoring of thermoluminescent dosimeters (TLDs) worn for a month, and with a questionnaire survey including habits of wearing protective devices and workloads. Each participant was contacted individually to schedule the monitoring program during the period when the participant could work for a whole month without vacation, between July and September 2017; this period coincided with the time at which the KCDC NDR-reporting for the 3rd quarter was scheduled, for 2017. According to the individual schedule, we sent the monitoring program package to participants, which included instructions pertaining to the study process, the TLD badges, and the questionnaire. Effort was made in the study design to optimize the complete participation rate by performing pre-evaluation and revision of the monitoring protocol, which ensured the convenience of the participants during the entire study process. All participants fully understood the monitoring protocol of the two dosimetry system and conformed to the instructions for wearing two TLD badges. Additionally, we sent text messages once a week to encourage and check for continued participation.

The TLD badges (UD-802 model, Panasonic, Secaucus, NJ, USA), which generally used for reporting doses of diagnostic radiation workers to NDR, were worn under the lead apron at the left chest, in accordance with the mandatory protocol issued by the KCDC managing NDR. Participants wore the TLD badge and recorded the start and completion date of the study, and the number of duty days. The TLD badge and questionnaire were collected via courier after the study period. TLD badge readings were obtained by one of the accredited centers (Orbitech, Seoul, Korea) designated by the KCDC. Doses were reported as the personal dose equivalent at a depth of 10 mm (Hp (10)) by the reading system (UD-716AGL TLD Readers, Panasonic, Secaucus, NJ, USA), based on the standard method recommended by the NDR in accordance with the Regulations for Safety Management of Diagnostic Radiation [17]. The minimum detectable level (MDL) was 0.01 mSv. The reported doses were corrected for background radiation in the same way as were the KCDC NDR doses.

### 2.3. Linkage with National Dose Registry Data

In South Korea, occupational radiation exposure of diagnostic radiation workers has been managed through the National Management System by the KCDC in accordance with the Rules for Safety Management of Diagnostic Radiation Emitting Generators [18] since 1996. The NDR of KCDC has collected the information of registered diagnostic radiation workers including interventional medical workers who are required to wear a TLD badge beneath the apron at left side of the chest mandatorily. The following information is collected by the KCDC: workers’ name, sex, date of birth, personal identification number, job classification, hospital name and address, quarterly reported dose data, and the start and end of the period of measurement. Dose data were reported quarterly by five centers for the personal dosimetry service designated by the KCDC. These data were based on measurements from the individual TLD badges issued to the monitored workers [19].

We requested the NDR data for the study participants from the KCDC, based on the availability of informed consent for the use of personal information. Each participant’s information was linked to the NDR database by matching the name, sex, and date of birth. Data linking was confirmed by job classification, and by the name and address of the hospital at which the participant worked.

### 2.4. Statistical Analysis

We compared actively monitored doses and the KCDC NDR doses for the third quarter in 2017 for the validity analyses. Most of the monitored doses were measured for one month, but 5 participants wore the badges for less than or more than one month. Monitored doses of these cases were adjusted to those corresponding to one month based on the actual number of days measured, and designated as monthly doses. If the monthly doses were the MDL, calculated quarterly doses were also assigned as the MDL. The MDLs were designated as 0.005 mSv for the statistical analysis as per a previous study [20].

The actively monitored doses were projected to obtain quarterly doses which were regarded as true doses. We conducted descriptive analyses of the participant’s characteristics and doses, and validity analyses of NDR doses relative to the projected quarterly monitored doses. The validity was assessed by Spearman’s correlation, the intraclass correlation coefficient (ICC), and the paired *t*-test [21].

Subgroup analyses were conducted based on whether the KCDC personal badges were worn regularly or not during interventional procedures. In the questionnaire, participants were asked how much time (100%, 75–99%, 25–74%, 1–24%, 0%) they wore the KCDC personal badge, usually during their practice. According to the answers, the regular badge wearers were defined as those who wore a badge for 75% or more of the time during their practice, and others were defined as irregular badge wearers. We constructed a modified Bland–Altman plot to compare the difference between the two doses against the actively monitored doses as a gold standard [22]. Dose differences between the actively monitored doses and the NDR doses in both groups were compared using the *t*-test with an unequal variance assumption.

Additional stratified analyses were performed based on five categories of level of wearing the KCDC badges according to the original question. To verify other factors related to agreement among regular badge wearers, Spearman’s correlation coefficients were calculated by job characteristics such as the year they started working, number of years worked, and wearing of protective devices; the difference in correlation coefficients was tested using the CORTESTI command in the STATA (version 14, College Station, TX, USA) software [23].

## 3. Results

Demographic and occupational characteristics of the study participants collected via the questionnaire survey are presented in Table 1. The majority of participants were men (92.9%) and the mean age was 44 years (standard deviation (SD), 7.3). Half of the participants began work as interventional radiologists after 2007 and the average duration of experience was 11 years (SD, 7.7). There were 37 regular badge wearers (66.1%) and 19 irregular badge wearers. The characteristics of regular badge wearers were generally similar to those of the irregular badge wearers.

Dosimetry data for the participants are summarized in Table 2. For all the participants, actively monitored doses which were projected to quarterly doses ranged from MDL to 15.07 mSv (mean 1.48 mSv), and the overall mean the KCDC NDR dose was 0.50 mSv. The mean monitored dose of irregular badge wearers (2.52 mSv) was significantly higher than that of regular badge wearers (0.95 mSv). In contrast, the mean NDR dose of regular badge wearers (0.69 mSv) was significantly higher than that of irregular badge wearers (0.13 mSv).

The correlation between actively monitored doses and the KCDC NDR doses was not observed in the entire study participants (ρ = 0.06) and the mean actively monitored dose was significantly higher than the mean NDR dose (mean difference 0.98 mSv, *p* = 0.008). In subgroup analyses, a significant correlation was observed between actively monitored doses and NDR doses in regular badge wearers (ρ = 0.43, *p* = 0.009). For irregular badge wearers, actively monitored doses and NDR doses showed a negative correlation (ρ = −0.21, *p* = 0.338). Estimation of the ICC showed similar results. No apparent difference was seen between actively monitored doses and NDR doses among regular badge wearers (mean difference 0.26 mSv, *p* = 0.272); however, the difference was significant among irregular badge wearers (mean difference 2.39 mSv, *p* = 0.013).

Figure 2 shows a modified Bland-Altman plot. The differences between actively monitored doses and NDR doses were plotted against actively monitored doses as reference values. A greater underestimation of NDR doses was observed as the actively monitored doses increased in both groups. The average dose difference between actively monitored doses and NDR doses in irregular badge wearers was significantly higher than that in regular badge wearers (*p* = 0.03).

Stratified analyses by categories of level of wearing the KCDC NDR badges presented a similar pattern for each category (Appendix A). Among participants who wore NDR badges regularly, there was no job characteristics which showed a significant difference in the correlation (Appendix A).

## 4. Discussion

Our findings indicated the low correlation between the periodically reported KCDC NDR doses and the actively monitored doses among Korean interventional radiologists, which may reflect an underestimation within NDR data. However, interventional radiologists who wore badges regularly presented with a higher correlation and a relatively smaller difference was observed between the NDR doses and actively monitored doses for this group. Thus, compliance information is important for epidemiological studies using reported doses.

The monthly mean dose measured for interventional radiologists in this study was 0.54 mSv (SD 0.95 mSv), i.e., one third of monitored quarterly dose (mean 1.48 mSv), which was higher than the average dose from eight Dutch interventional radiologists (mean 0.12 mSv, SD 0.11 mSv) [24] and 29 US interventional radiologists (mean 0.09 mSv) [25]. This difference may be due to the increased number of cases of interventional procedures compared to the previous studies and different position of the monitored badge in the US study; in the US study the badges were to be worn beneath the apron at waist level. However, the median annual dose extrapolated from quarterly NDR reported doses was 0.76 mSv, which was similar with those of an international study for interventional cardiologists from 23 countries (i.e., 0.73 mSv) based on reported doses in 2008 [26].

These findings suggest that the reported NDR doses were underestimated compared to the actually measured doses in this study. Underestimation of true doses could lead to an overestimation of the dose-response relationship [27]. Simulation studies from Canada [28] and the US [29] showed a 6–20% overestimation in risk estimates from underestimated doses induced by censoring of MDL doses although this did not affect the properties of the statistical significance test for risk estimates. The influence on risk estimation needs to be evaluated in each epidemiological study, because such risks could be different at different levels of underestimation of exposure and may also be influenced by a combination of various measurement errors.

Furthermore, irregular badge wearers showed higher actually monitored doses than did regular badge wearers. The difference between the actively monitored doses and the NDR doses among irregular wearers was 10 times that among regular wearers. This would imply that interventional radiologists who were actually more exposed to radiation showed lower compliance with badge-wearing policies, which may have led to lower NDR doses. Interventionalists including radiologists, cardiologists, and surgeons may have a tendency to accept their risk as inherent in the profession and may be concerned about being excluded from procedures due to exceeded doses [30]; this aspect may have aggravated the low compliance rate. Because some interventional medical radiation workers did not wear personal badges routinely in South Korea [31], Europe [32], and in the US [33], considerable attention is required to interpret and utilize NDR data based on compliance of radiation medical workers with badge-wearing policies.

We also need to examine the specific reasons for the low compliance rate of wearing a badge among irregular wearers. Low compliance of badge wearing could increase the unmonitored dose and miss the undetected highly exposed group. The customized education and training for irregular badge wearers are, therefore, needed to encourage them to wear the badges strictly. In addition, although regulatory authority has controlled radiation workers who exceed recommended dose levels in South Korea, additional monitoring and inspection should be paid to repeatedly reported doses under the minimum.

There are several limitations to the study. First, we measured the actual doses for one month to avoid loss to follow-up, however, the average number of procedures performed during the remaining two months of the quarter may be different from the number performed during the monitored month. In addition, the complexity of the procedures or characteristics of patients may affect the dose even if the number of procedures remains the same [13]. Additionally, we assigned the MDL dose as 0.005 mSv for the projected quarterly doses for participants who received the MDL during the monitored month (*n* = 9). If we had monitored for additional two months, these participants may have received higher monitored doses than the MDL and the difference between actively monitored doses and NDR doses among regular wearers may be smaller. However, the one-month was practically feasible maximum monitoring period based on the prior discussion with the staff of KSIR and pilot study participants. The longer period of monitoring would have led to a high rate of loss to follow-up. To obtain accurately monitored doses, we closely followed-up participants with consistent instructions, which could have contributed to minimizing measurement errors.

Second, it was not possible that two dosimeters were placed at exactly the same location. Slightly different dosimetric locations, such as the left eye and between the eyes, resulted in inconsistent doses for the same procedure [34]. This might produce not having completely identical values from two different badges even in the same monitoring condition.

Third, this investigation includes the small number of study participants; however, the enrolled participants comprised 21.5% of active members of the KSIR, which is relatively representative of the target population. Depending on the job titles or specialties, the validity of NDR data could be varied which is basically related with the compliance of wearing a badge but it is also affected by education, training and safety culture [35]; for this reason, our results are restricted to the specific population, which may limit generalizability to the entire NDR data.

## 5. Conclusions

Our study provided the information of occupational radiation doses and indicated the underestimation of the NDR data among interventional radiologists in South Korea. Because the validity of NDR was significantly dependent on compliance, collecting the information regarding badge-wearing compliance would be important for epidemiological studies using reported doses. Furthermore, considerable effort is required to interpret and utilize the NDR data, and further studies are needed to evaluate the influence of the dose uncertainty induced by underestimated true dose resulting from noncompliance.

## Figures and Tables

**Figure 1 ijerph-18-04195-f001:**
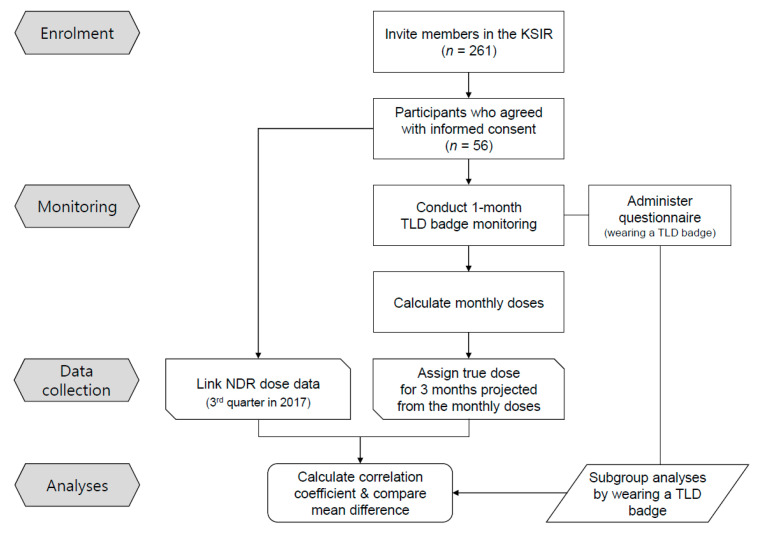
Study flowchart. KSIR, Korean Society of Interventional Radiology; TLD, thermoluminescent dosimeter; NDR, national dose registry.

**Figure 2 ijerph-18-04195-f002:**
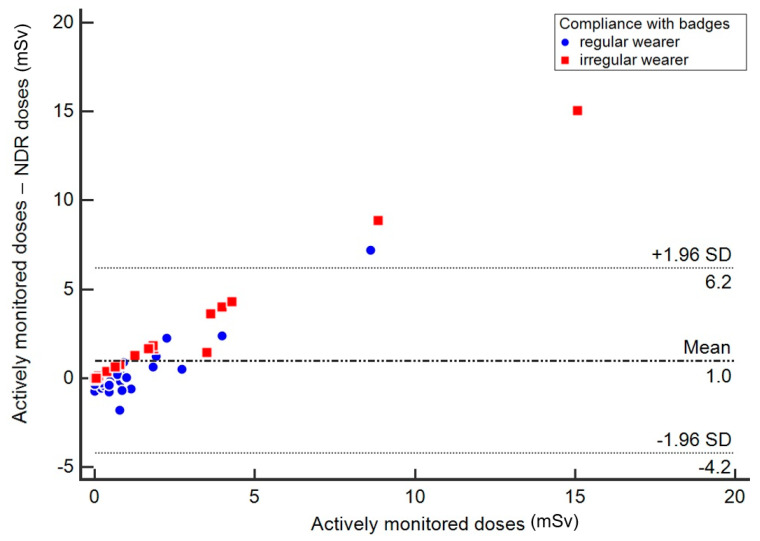
Modified Bland–Altman plot of the difference between actively monitored doses and NDR (national dose registry) doses. Regular wearers are defined as those wearing the KCDC (Korea Centers for Disease Control and Prevention) NDR badge for 75% or more of the time during their practice, and irregular wearers are defined as those wearing the KCDC NDR badge for less than 75% of the time during their practice.

**Table 1 ijerph-18-04195-t001:** Characteristics of the study participants by wearing the KCDC (Korea Centers for Disease Control and Prevention) badges.

Characteristics ^1^	Total Participants (*N* = 56)		Participants Who Wore a Badge for ≥75% of the Time (*N* = 37)		Participants Who Wore a Badge for <75% of the Time (*N* = 19)		*p* Value ^2^
	N	(%)	N	(%)	N	(%)	
Sex							0.696
Male	52	(92.9)	34	(91.9)	18	(94.7)	
Female	4	(7.1)	3	(8.1)	1	(5.3)	
Age							0.784
<45 years	31	(55.4)	20	(54.1)	11	(57.9)	
≥45 years	25	(44.6)	17	(45.9)	8	(42.1)	
Calendar year began working as an interventional radiologist							0.778
<2007	28	(50.0)	19	(51.4)	9	(47.4)	
≥2007	28	(50.0)	18	(48.6)	10	(52.6)	
Years worked as an interventional radiologist							0.505
<10 years	26	(46.4)	16	(43.2)	10	(52.6)	
≥10 years	30	(53.6)	21	(56.8)	9	(47.4)	
Wearing a lead apron							0.470
100%	55	(100.0)	36	(100.0)	19	(100.0)	
<100%	0	(0.0)	0	(0.0)	0	(0.0)	
Wearing a thyroid shield							0.229
100%	52	(94.5)	35	(97.2)	17	(89.5)	
<100%	3	(5.5)	1	(2.8)	2	(10.5)	
Using ceiling-suspended shielding							0.957
≥75%	20	(36.4)	13	(36.1)	7	(36.8)	
<75%	35	(64.6)	23	(63.9)	12	(63.2)	
Using table-suspended shielding							0.817
≥75%	33	(60.0)	22	(61.1)	11	(57.9)	
<75%	22	(40.0)	14	(38.9)	8	(42.1)	

^1^ Numbers may not add up to total due to missing information; ^2^
*p* value obtained from the chi-square test for categorical variables and the Student *t*-test for continuous variables.

**Table 2 ijerph-18-04195-t002:** Comparison of actively monitored doses and the KCDC (Korea Centers for Disease Control and Prevention) NDR (national dose registry) doses (Hp (10)) for interventional radiologists during a quarter in 2017.

	Total Participants (*n* = 56)				Participants Who Wore a Badgefor ≥75% of the Time (*n* = 37)				Participants Who Wore a Badge for <75% of the Time (*n* = 19)				*p* Value ^1^
	Mean	(SD)	Median	(Range)	Mean	(SD)	Median	(Range)	Mean	(SD)	Median	(Range)	
Quarterly doses (mSv)													
Actively monitored doses	1.48	(2.61)	0.74	(0.005–15.07)	0.95	(1.56)	0.47	(0.005–8.61)	2.52	(3.78)	1.27	(0.005–15.07)	0.032
NDR doses	0.50	(0.65)	0.19	(0.005–2.56)	0.69	(0.65)	0.65	(0.005–2.56)	0.13	(0.47)	0.005	(0.005–2.07)	0.002
Validation analyses													
Spearman’s correlation Coefficient (*p* value)	0.058 (0.672)				0.425 (0.009)				−0.210 (0.388)				
ICC ^2^ (*p* value)	0.049 (0.419)				0.451 (0.037)				0.017 (0.048)				
Dose difference (mean ± SD)	0.98 ± 2.65				0.26 ± 1.42				2.39 ± 3.78				
Paired *t* test	0.008				0.272				0.013				

^1^* p* value obtained from the Student *t*-test assuming unequal variance between mean doses of participants who have worn a badge 75% or more and less than 75% of the time; ^2^ ICC, intraclass correlation coefficient.

## Data Availability

Access to detailed individual data is restricted for both legal and ethical concerns. The availability of data should be reviewed with the corresponding author and approved by the Institutional Review Board of Korea University.

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
