# Peer review of "Occupational Radiation Exposure and Validity of National Dosimetry Registry among Korean Interventional Radiologists"

_ijerph, 2021, doi:10.3390/ijerph18084195_

Round 1
Reviewer 1 Report
What was the reason for low correlation between NDR and active dose monitoring badge results. It seems that wearing the badge makes a difference. What is the significance of that?
Author Response
Point 1: What was the reason for low correlation between NDR and active dose monitoring badge results. It seems that wearing the badge makes a difference. What is the significance of that?

Response 1: Yes. We have believed that low correlation came from difference of wearing the badge. Accuracy of the NDR data is essential to protect radiation workers and epidemiological studies of low dose and low dose rate radiation exposure. There are various sources of inaccuracy, however, the direct validation of reported doses has not been considered so far. Compliance with badge-wearing policies is not guaranteed for individual workers, which leads to unreliable dose data.
Our study showed low correlation and potentially a large difference between the NDR doses and the actively monitored doses among Korean interventional radiologists, which may reflect an underestimation within NDR data. A greater underestimation of NDR doses was observed as the actively monitored doses increased in regular wearers and irregular wearers. According to these results, badge wearing habits could affect to the uncertainty of NDR dose data and the regulatory strategies. Therefore, collecting information of the badge in epidemiological studies using reported dose data. For the safety management of radiation workers, these are needed that detecting irregular badge wearers, training the specific target, and monitoring the repeatedly reported very low doses.
We have added the significance of validity of NDR data and the factor influencing validity in the Introduction (line 49-50).
‘Knowledge on validity of NDR data and furthermore factors influencing validity could contribute to reduce uncertainty of the reported dose data.’
The detailed approaches are needed further studies and we have added the significance of wearing the badge in view of the management of radiation medical worker points in the Discussion (line 258-265).
‘In addition, we need to examine the specific reason for the low compliance rate among irregular badge wearers for the effective management of noncompliance. The customized education and training is also needed for irregular badge wearers to encourage them to wear the badges strictly. The worst case scenario might be that unmanaged noncompliance of the badge leads to miss the undetected highly exposed group. Regulatory authority has controlled radiation workers who exceed recommended dose levels in South Korea, however, a specific attention and inspection should be paid to repeatedly reported doses under the minimum.’
Reviewer 2 Report
This study describes an important area of occupational health, involving active monitoring compared with reported Registry dose for a potential high exposure/risk group of intervention radiologists. This was coupled with a questionnaire for time/task/activity patterns of work. It describes the validity of reported doses, and the importance of compliance with monitoring protocols, particularly for epidemiologists.
The Methods are clearly described, and accessible to general scientific audience, especially those with an occupational health interest.
The Results clearly demonstrate that monitored dose was higher for this risk group than NDR estimates. This was further influenced by compliance with badge wearing protocols. This study is a Good example of how to identify and target education/training priorities to reduce exposure in high risk workers. The Results were interpreted clearly, and for different stakeholders (e.g. workers and epidemiologists). Limitations were discussed and are acceptable to this reviewer.
Some recommendations for further work could be included, such as education/training and worker management.
There are some minor English grammatical errors throughout e.g.
-
L54-5 unclear English “Moreover, ...... it is concerned that radiation risk to interventional medical workers”. Revise.
- L154 change ‘was’ to ‘were’
-
L213 remove the last ‘the’.
Author Response
Point 1: This study describes an important area of occupational health, involving active monitoring compared with reported Registry dose for a potential high exposure/risk group of intervention radiologists. This was coupled with a questionnaire for time/task/activity patterns of work. It describes the validity of reported doses, and the importance of compliance with monitoring protocols, particularly for epidemiologists.
The Methods are clearly described, and accessible to general scientific audience, especially those with an occupational health interest.
The Results clearly demonstrate that monitored dose was higher for this risk group than NDR estimates. This was further influenced by compliance with badge wearing protocols. This study is a Good example of how to identify and target education/training priorities to reduce exposure in high risk workers. The Results were interpreted clearly, and for different stakeholders (e.g. workers and epidemiologists). Limitations were discussed and are acceptable to this reviewer.
Some recommendations for further work could be included, such as education/training and worker management.
Response 1: Thank you for the comment. We added suggested points in the Discussion (line 258-265).
‘In addition, we need to examine the specific reason for the low compliance rate among irregular badge wearers for the effective management of noncompliance. The customized education and training is also needed for irregular badge wearers to encourage them to wear the badges strictly. The worst case scenario might be that unmanaged noncompliance of the badge leads to miss the undetected highly exposed group. Regulatory authority has controlled radiation workers who exceed recommended dose levels in South Korea, however, a specific attention and inspection should be paid to repeatedly reported doses under the minimum.’
Point 2: There are some minor English grammatical errors throughout e.g.
- L54-5 unclear English “Moreover, ...... it is concerned that radiation risk to interventional medical workers”. Revise.
- L154 change ‘was’ to ‘were’
- L213 remove the last ‘the’.
Response 2:
- To clarify the sentence, we have changed the sentence, ‘Moreover, according to the increase of these procedures, concerns are mounting about radiation exposure risk to interventional medical workers’ according to the reviewer`s comment (line 56-57).
- We have changed the word, ‘was’ to ‘were’ (line 162).
- We have removed the last ‘the’ in the line 213 (line 220).
Reviewer 3 Report
The authors investigated discrepancy of radiation doses between national registry and active monitoring in radiologists of Korea. It is important warning for regulatory bodies and also publics. But, there are several comments to improve the presentation and improvement of interpretation.
The discrepancy came from exact period of wearing dosimeters. Authors obtained information of wearing time at 5-point scale. Radiation doses should be shown in each point to elucidate the importance of compliance.
In addition, the time of wearing dosimeter determines the actual monitoring period. Then the dose in quarterly can be corrected by the actual wearing time. Wearing time corrected dose can be provided to elucidate potential effect of the time and importance.
Absolute difference in doses was affected by low dose data. Ratios between active monitoring and NDR dose can be provided and with plot against active monitoring dose like as Fig. 2.
To eliminate low dose data near detection limit, authors provide stratified analysis that uses data of dose more than 0.05 or 0.1 mSv.
Participants wore two dosimeters. How did authors control the compliance? Participants wore authors’ dosimeter completely, but not for NDR dosimeter? Authors should provide information on compliance for both dosimeters.
Line 111: Correct ‘medial’.
Round 2
Reviewer 1 Report
Why you did not include your validity analysis in your paper?
Author Response
We included our validity analysis in Table 2. As we described in the Methods section, we compared actively monitored doses and the national dose registry (NDR) doses reported to the Korea Centers for Disease Control and Prevention for the validity analyses. The actively monitored doses were regarded as reference standard measures. The validity was assessed by Spearman`s correlation, the intraclass correlation coefficient, and the paired t-test (please see Table 2 in the manuscript).
To clarify, we have revised the sentence as “We compared actively monitored doses and the KCDC NDR doses for the third quarter in 2017 for the validity analyses” in Materials and Methods (line 133).

Reviewer 3 Report
Now, I'm satisfied with the revision.
Author Response
We appreciate the reviewer’s comments on our manuscript.